# Validation of a new automated chemiluminescent anti-SARS-CoV-2 IgM and IgG antibody assay system detecting both N and S proteins in Japan

Rin Yokoyama[1], Makoto Kurano[1,2]*, Yoshifumi Morita[1], Takuya Shimura[1], Yuki Nakano[1], Chungen Qian[3], Fuzhen Xia[4], Fan He[4], Yoshiro Kishi[5], Jun Okada[5], Naoyuki Yoshikawa[1], Yutaka Nagura[6], Hitoshi Okazaki[6], Kyoji Moriya[7], Yasuyuki Seto[8], Tatsuhiko Kodama[9], Yutaka Yatomi[1,2]*

1 Department of Clinical Laboratory, The University of Tokyo Hospital, Tokyo, Japan, 2 Department of Clinical Laboratory Medicine, Graduate School of Medicine, The University of Tokyo, Tokyo, Japan, 3 The Key Laboratory for Biomedical Photonics of MOE at Wuhan National Laboratory for Optoelectronics—Hubei Bioinformatics & Molecular Imaging Key Laboratory, Systems Biology Theme, Department of Biomedical Engineering, College of Life Science and Technology, Huazhong University of Science and Technology, Hubei, P. R. China, 4 Reagent R&D Center, Shenzhen YHLO Biotech Co., Ltd, Guangdong, P. R. China, 5 Business Planning Department, Sales & Marketing Division, Medical & Biological Laboratories Co., Ltd, Tokyo, Japan, 6 Department of Blood Transfusion, The University of Tokyo Hospital, Tokyo, Japan, 7 Department of Infection Control and Prevention, The University of Tokyo, Tokyo, Japan, 8 Department of Gastrointestinal Surgery, The University of Tokyo, Tokyo, Japan, 9 Laboratory for Systems Biology and Medicine, The University of Tokyo, Tokyo, Japan

* kurano-tky@umin.ac.jp (MK); yatoyuta-tky@umin.ac.jp (YY)

**Data Availability Statement:** All relevant data are within the manuscript and its Supporting Information files.

## Abstract

PCR methods are presently the standard for the diagnosis of Coronavirus disease 2019 (COVID-19), but additional methodologies are needed to complement PCR methods, which have some limitations. Here, we validated and investigated the usefulness of measuring serum antibodies against severe acute respiratory syndrome coronavirus 2 (SARS-CoV-2) using the iFlash3000 CLIA analyzer. We measured IgM and IgG titers against SARS-CoV-2 in sera collected from 26 PCR-positive COVID-19 patients, 53 COVID-19-suspected but PCR-negative patients, and 20 and 100 randomly selected non-COVID-19 patients who visited our hospital in 2020 and 2017, respectively. The repeatability and within-laboratory precision were obviously good in validations, following to the CLSI document EP15-A3. Linearity was also considered good between 0.6 AU/mL and 112.7 AU/mL for SARS-CoV-2 IgM and between 3.2 AU/mL and 55.3 AU/mL for SARS-CoV-2 IgG, while the linearity curves plateaued above the upper measurement range. We also confirmed that the seroconversion and no-antibody titers were over the cutoff values in all 100 serum samples collected in 2017. These results indicate that this measurement system successfully detects SARS-CoV-2 IgM/IgG. We observed four false-positive cases in the IgM assay and no false-positive cases in the IgG assay when 111 serum samples known to contain autoantibodies were evaluated. The concordance rates of the antibody test with the PCR test were 98.1% for SARS-CoV-2 IgM and 100% for IgG among PCR-negative cases and 30.8% for SARS-CoV-2 IgM and 73.1% for SARS-CoV-2 IgG among PCR-positive cases. In

**Funding:** This work was conducted by the collaborative research project among The University of Tokyo, Shenzhen YHLO Biotech Co., Ltd, and Medical & Biological Laboratories Co. This work is also supported by the Research Grants in the Natural Sciences, The Mitsubishi Foundation awarded to M.K. F. X. and F. H. are employees of Shenzhen YHLO Biotech Co., Ltd and Y. K. and J. O. are employees of Medical & Biological Laboratories Co., Ltd.The funders provided support in the form of salaries for authors (F. X., F. H., Y. K., and J. O.), but did not have any additional role in the study design, data collection and analysis, decision to publish, or preparation of the manuscript. The specific roles of these authors are articulated in the 'author contributions' section.

**Competing interests:** The present study is a collaborative research project among The University of Tokyo, Shenzhen YHLO Biotech Co., Ltd, and Medical & Biological Laboratories Co., Ltd. F. X. and F. H. are employees of Shenzhen YHLO Biotech Co., Ltd and Y. K. and J. O. are employees of Medical & Biological Laboratories Co., Ltd. Other authors received no consultancy or patents from these companies. The SARS-CoV-2 IgM and IgG CLIA kits, which detect N and S proteins, were marketed products and the SARS-CoV-2 IgM and IgG CLIA kits, which detect N or S proteins separately, were the products in development. These conflicts of interest do not alter our adherence to PLOS ONE policies on sharing data and materials.

**Abbreviations:** ANOVA, analysis of variance; AU, arbitrary unit; CLIA, chemiluminescence immunoassay; CLSI, The Clinical & Laboratory Standards Institute; COI, cutoff index; COVID-19, coronavirus disease 2019; CV, coefficient variation; dsDNA, double-strand DNA; ECLIA, electrochemiluminescence immunoassay; FDA, the Food and Drug Administration; IgG, immunoglobulin G; IgM, immunoglobulin M; M2, mitochondrial M2; N protein, Nucleocapsid protein; P-ANCA, myeloperoxidase antineutrophil cytoplasmic antibody; RF, rheumatoid factor; RLU, relative light units; SARS-CoV-2, severe acute respiratory syndrome coronavirus 2; SD, standard division; S protein, Spike protein; SS-A, Sjögren's syndrome A.

conclusion, the performance of this new automated method for detecting antibody against both N and S proteins of SARS-CoV-2 is sufficient for use in laboratory testing.

## Introduction

In December 2019, the first pneumonia cases caused by an unknown microorganism were identified in Wuhan, China [1]. The pathogen was identified as a novel betacoronavirus and was named "severe acute respiratory syndrome coronavirus 2 (SARS-CoV-2)" [2]. SARS-CoV-2 is phylogenetically similar to SARS-CoV, which caused outbreaks of a severe respiratory syndrome in China in 2002 [3]. The symptoms of coronavirus disease 2019 (COVID-19), which is the respiratory syndrome caused by SARS-CoV-2, are fever, cough and lymphopenia [4]. Chest computed tomography examinations of COVID-19 patients are characterized by the bilateral distribution of patchy shadows or ground-glass opacities [5]. Since early December 2019 and as of June 15, 2020, over 7,800,000 cases of COVID-19 have been confirmed and 430,000 deaths have been reported throughout the world [6], and the World Health Organization has reported a fatality rate for cases defined as pneumonia of approximately 2% [7].

Currently, COVID-19 is diagnosed by the clinical presentation of the patient, as described above, and the detection of SARS-CoV-2 RNA in respiratory specimens, such as a nasal swab or sputum, using real-time reverse-transcription polymerase chain reaction (RT-PCR) [8, 9]. However, this method requires skilled technicians who know how to handle genetic samples and perform PCR tests and occasionally causes false-negative results because of the viral replication window, a low viral titer, or incorrect sample collection [10]. Moreover, the sampling of respiratory specimens exposes medical staff to a higher risk of secondary infection through aerosolization than the sampling of sera [11, 12]. Therefore, other methods are required to complement PCR testing.

Candidate complementary tests include both antigen and antibody tests. Regarding antigen tests, although this method does not require special skills for performing genetic testing, there remains a high risk of secondary infection during sampling, and the sensitivity of antigen tests is reportedly lower than that of PCR testing [13, 14]. Antibody tests are another candidate. Compared with PCR testing, this serological test method has a faster turn-around time and requires easier and safer sample collection and less specialized skills for technicians; furthermore, when we interpret the results of an antibody testing considering the duration after the onset of COVID-19, this test would give us important information in diagnosing COVID-19. The main concern regarding antibody tests is the high frequency of false-positive cases, which is a parameter that should depend on the quality of the test product [15, 16]. Recently, Shenzhen YHLO Biotech Co., Ltd (China) has developed an antibody test with a high specificity [17–19]; however, this method has not yet been validated in the Japanese population. Therefore, in the present study, we aimed to validate the measurement of IgM and IgG antibodies against SARS-CoV-2 in sera and to investigate the usefulness of this method for the diagnosis of COVID-19.

## Materials and methods

### Subjects

We enrolled a total of 26 COVID-19-positive cases and 53 COVID-29-suspected cases who were hospitalized at The University of Tokyo Hospital. Confirmed COVID-19 cases were defined based on the guidelines of The University of Tokyo Hospital. Briefly, patients with

acute respiratory infection syndrome accompanied by detectable SARS-CoV-2 RNA in a throat swab or sputum at least once were confirmed as having COVID-19 (PCR-positive cases). Suspected patients were defined as subjects with respiratory symptoms, a history of overseas travel, or a high-risk contact with a confirmed COVID-19 case but with negative PCR results. We received the patient's whole blood in the collection tube coated with silica and thrombin for the clinical laboratory testing. Then, the serum was separated by centrifuging at 2,300 g for 5 minutes and carried out clinical testing. We collected residual sera available after routine clinical testing and kept it frozen at -20˚C until measurement. The serum samples before the onset of the infection were collected by chance since all serum samples investigated in our laboratory were stored for 3 weeks from the day when the routine laboratory testing was performed. Three of the subjects of whom we collected the serum samples before the onset of COVID-19 symptoms had been confirmed SARS-CoV-2 PCR negative. In other four subjects and control cases collected in March 2020, symptoms of cold were not described in medical records. We enrolled the COVID-19 subjects and the suspicious subjects, which we can collect residual serum samples between April 22, 2020 and June 22, 2020. The serological tests were performed using the sample that had been collected on the day closest to the day on which the sample for the PCR test had been collected. The mean days (±S.D.) between the antibody test and the onset of the symptom or the PCR test were 11.3 (±6.70) or 5.67 (±5.67) days, respectively. As control groups, we randomly selected 20 and 100 outpatients who had visited The University of Tokyo Hospital in March 2020 or January-December 2017 and were not complicated with autoimmune diseases, respectively. We also collected 111 serum samples known to contain autoantibodies. All samples used in this study were stored biological samples and were de-identified before we accessed them.

The current study was performed in accordance with the Declaration of Helsinki. Informed consent was obtained in the form of an opt-out form on the institution's website. The institutional ethics committee approved this informed consent plan. This study was conducted with the approval of The University of Tokyo Medical Research Center Ethics Committee (2019300NI-3).

## Antibody testing

Antibody testing was performed using SARS-CoV-2 IgM and IgG chemiluminescence immunoassay (CLIA) kits supplied by Shenzhen YHLO Biotech Co., Ltd. (China) and an iFlash3000 fully automated CLIA analyzer also from Shenzhen YHLO Biotech Co., Ltd. (China). Two antigens of SARS-CoV-2 (nucleocapsid protein [N protein] and spike protein [S protein]) were coated on the magnetic beads of these CLIA assays. The assay procedure was described by Qian C, et al [20]. Briefly, the acridinium-labeled anti-human IgM or IgG conjugate antibody was used to detect the antibody bound to the beads. The SARS-CoV-2 IgM or IgG titers in 5 uL of the sample were calculated as relative light units (RLU) obtained from the analyzer and were described as arbitrary units per milliliter (AU/mL) by comparing the RLU detected by iFlash optical system with the cutoff calculated from the SARS-CoV-2 IgM or IgG calibrators containing anti-SARS-CoV-2 IgM or IgG chimeric antibody. According to the manufacturer's instructions, the cutoff value for a positive SARS-CoV-2 IgM/IgG result was deemed as 10 AU/mL. To measure the antibody titers against a single antigen, we used the magnetic beads coated with either antigen. If the SARS-CoV-2 IgG titer was over 40 AU/mL, the sample was diluted with non-reactive serum and the antibody titers were measured once again.

To compare with an established antibody test against SARS-CoV-2, we measured 385 serum samples collected from COVID-19 positive or suspected patients with an anti-SARS-CoV-2 electrochemiluminescence immunoassay (ECLIA) kit obtained from Roche diagnostics

K.K. (Japan), equipped with cobas e 601 manufactured by Roche diagnostics K.K. (Japan). According to the manufacturer's instructions, the cutoff value for a positive SARS-CoV-2 antibody result was deemed as 1 COI (cutoff index).

## Method validation

We evaluated the assay precision, according to the guideline of The Clinical & Laboratory Standards Institute (CLSI) documents EP15-A3 [21]. We investigated the assay precision for five days and five replicates per run, using two manufacturer's controls as a sample. We compared the repeatability and within-laboratory precision, using the results obtained in this study and the upper verification limits calculated from the precision values proposed by Qian C et al, which were obtained by measuring three to four serum samples in duplicates for each run and two runs per day over 20 testing days, following CLSI documents EP5-A3 [20]. Linearity was investigated using two kinds of pooled serum samples. Briefly, each sample was diluted with pooled non-reactive serum in two-fold serial dilutions for ten times. Additionally, linearity study was also performed according to the CLSI document EP06 [22]. The range which any nonlinear coefficients were not significant when a least-squared regression using polynomials of 1st, 2nd and 3rd order was regarded as a linear range. To investigate the existence of the prozone phenomenon, we diluted the samples with high concentrations of SARS-CoV-2 IgM and IgG titers using ten-fold serial dilutions for the SARS-CoV-2 IgM assay and two-fold serial dilutions for the SARS-CoV-2 IgG assay.

To evaluate the detection capability, we performed the verification of limit of blank (LoB), limit of detection (LoD), and limit of quantitation (LoQ), according to the CLSI document EP17-A2 [23]. The LoD was determined with the proportions of false positives (α) less than 5% and false negatives (β) less than 5%. Non-reactive serum was used as a blank sample. To evaluate the LoQ, we measured 8 samples for SARS-CoV-2 IgM and 10 samples for SARS-Cov-2 IgG with 6 replicates, respectively. The LoQ was determined as the minimum concentration no more than 10% of CV. We performed method validation using a single reagent lot.

## Statistical analysis

For the precision study, analysis of variance (ANOVA) was used to estimate the repeatability and within-laboratory precision. For linearity study, whether a nonlinear coefficient was significantly different from zero was evaluated by t-test. These data were analyzed using JMP software (North Carolina, USA). The data evaluating the cross-reactivity were analyzed using StatFlex software (Osaka, Japan). The results were expressed as the mean ± SD. The Dunn test was used for comparisons of antibody titers between the control and other groups. A value of $p < 0.05$ was regarded as statistically significant in all the analyses.

## Results

### Precision and accuracy of antibody testing

First, we performed the verification of repeatability and within-laboratory precision according to the CLSI document EP15-A3 for five days and with five replicates per run, using two manufacturer's controls each assay (Table 1). The repeatability of SARS-CoV-2 IgM was ranged from 1.90% to 2.13%, and the within-laboratory precision was from 2.48% to 4.08%. The repeatability of SARS-CoV-2 IgG was ranged from 1.03% to 1.65%, and the within-laboratory precision is from 1.87% to 2.17% (Table 1). Those precision values were lower than the upper verification limits (from 2.80% to 4.32 for the repeatability of SARS-CoV-2 IgM, from 3.02% to 4.08% for the within-laboratory precision of SARS-CoV-2 IgM, from 3.11 to 4.30% for the

**Table 1. Repeatability and within-laboratory precision of SARS-CoV-2 IgM/IgG according to CLSI EP15-A3.**

| | Sample | Mean (AU/mL) | Repeatability | | Within-laboratory precision | |
|---|---|---|---|---|---|---|
| | | | SD | CV (%) | SD | CV (%) |
| SARS-CoV-2 IgM | 1 | 5.4 | 0.12 | 2.13 | 0.22 | 4.08 |
| | 2 | 25.2 | 0.48 | 1.90 | 0.63 | 2.48 |
| SARS-CoV-2 IgG | 3 | 5.5 | 0.09 | 1.65 | 0.10 | 1.87 |
| | 4 | 19.6 | 0.20 | 1.03 | 0.43 | 2.17 |

We measured SARS-CoV-2 IgM or IgG in two manufacturer's controls for 5 days, 5 replicates per run, following to the CLSI document EP15-A3. Repeatability and within-laboratory precision were calculated using ANOVA.

repeatability of SARS-CoV-2 IgG, and from 3.12 to 5.13% for the within-laboratory precision of SARS-CoV-2 IgG), which were calculated from the precision values reported by Qian C et al [20].

**Measurement range of SARS-CoV-2 antibody testing.** To explore the measurement range of this antibody test, we performed a linear regression analysis. When we investigated linearity using samples with moderate antibody titers, the curves showed a good linearity between 0.6 AU/mL and 112.7 AU/mL for SARS-CoV-2 IgM and between 3.2 AU/mL and 55.3 AU/mL for SARS-CoV-2 IgG (Fig 1A–1D). Additionally, none of nonlinear coefficients were significant between 1.50 AU/mL and 15.92 AU/mL for SARS-CoV-2 IgM ($p = 0.66$ for $2^{nd}$ order regression, $p = 0.50$ and $p = 0.47$ for $3^{rd}$ order regression) and between 2.36 AU/mL and 18.30 AU/mL for SARS-CoV-2 IgG ($p = 0.27$ for $2^{nd}$ order regression, $p = 0.13$ and $p = 0.10$ for $3^{rd}$ order regression) (Fig 1E and 1F). Next, we measured samples with high antibody titers to determine the upper limit of the measurement range. In the SARS-CoV-2 IgM assay, the upper curve increased up to a value of 2,405 AU/mL and then reached a plateau at higher concentrations. In the SARS-CoV-2 IgG assay, the curve reached a plateau at values over 73 AU/mL (Fig 2A and 2B). When we used saline as a diluent, we observed a hook effect (Fig 2C), whereas when we used non-reactive serum, we did not observe a hook effect as shown in Fig 2A and 2B. Therefore, we diluted the samples over 40 AU/mL for SARS-CoV-2 IgG with non-reactive serum.

**Detection capability of SARS-CoV-2 antibody testing.** To evaluate the detection capacity for SARS-CoV-2 IgM/IgG, we determined the LoB and LoD. The LoD for SARS-CoV-2 IgM was 0.74 AU/mL, determined by 130 measurements with 60 blank and 70 low level replicates, and the LoB was 0.63 AU/mL. The LoD for SARS-CoV-2 IgG was 0.53 AU/mL, determined by 120 measurements with 60 blank and 60 low level replicates, and the LoB was 0.47 AU/mL. We also investigated the LoQ. We measured 7 samples of low antibody levels for SARS-CoV-2 IgM and 10 samples for SARS-CoV-2 IgG. In both assays, CVs of the samples which were lower than the LoD showed no more than 10%. Therefore, the LoQ for SARS-CoV-2 IgM was determined as 0.74 AU/mL and that for SARS-CoV-2 IgG was 0.53 AU/mL.

**Successful detection of SARS-CoV-2 IgM/IgG.** To confirm that this antibody measurement system could detect SARS-CoV-2 IgM/IgG successfully, we measured the antibody titers in sera obtained before and after infection with SARS-CoV-2 in three cases of COVID-19 confirmed using RT-PCR tests. As shown in Fig 3, SARS-CoV-2 IgM and IgG were not detected before symptom onset; at several days after symptom onset, tests for both antibodies became positive and the titers gradually increased. In case 4, 6 and 7, SARS-CoV-2 IgM titers gradually increased although the titers did not increase above the cutoff value in the period of observation. In case 1 and 3, the IgM test became negative again at day 19 and day 22, respectively.

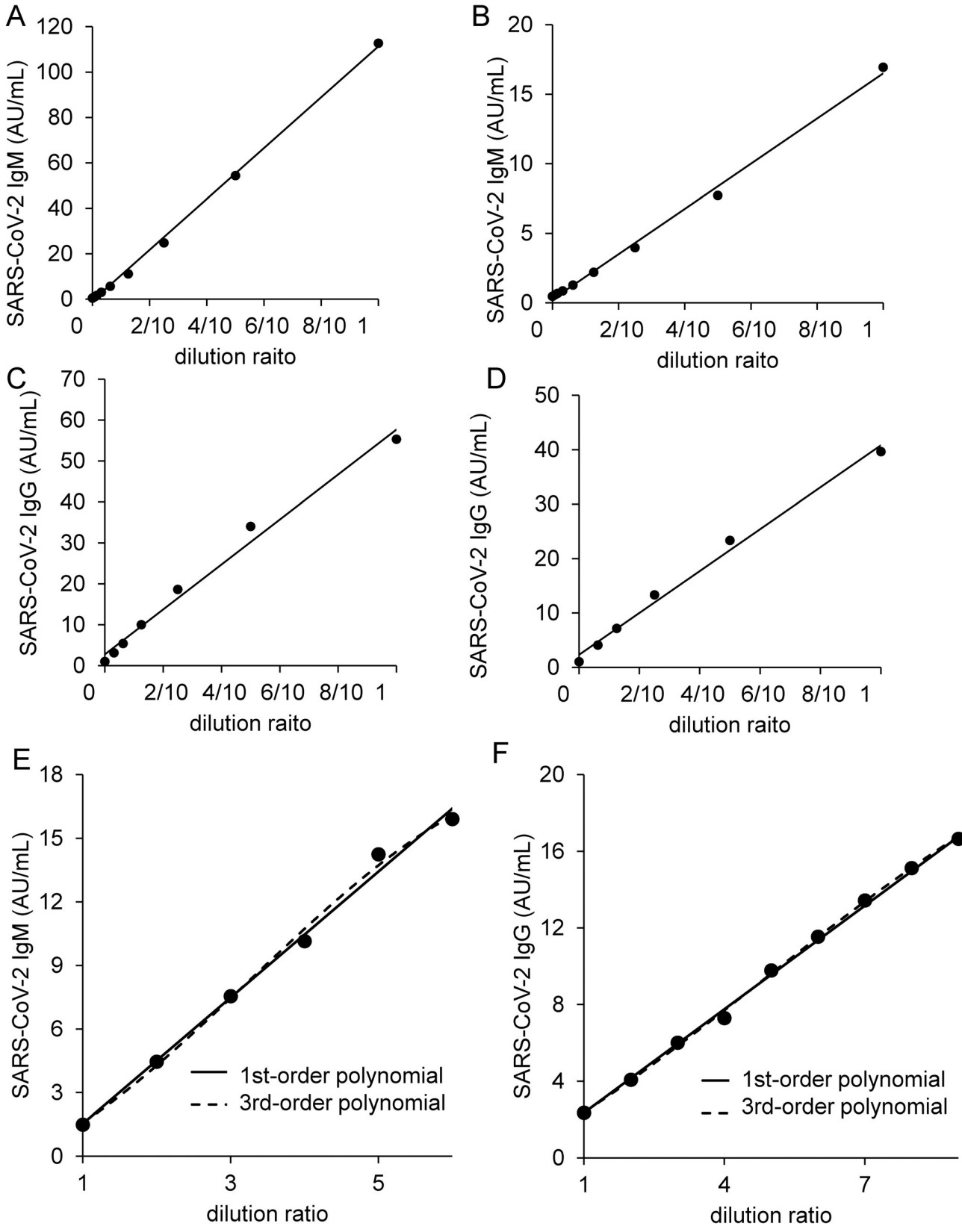

**Fig 1. Linearity analyses of SARS-CoV-2 antibody titer.** The dilution linearities of SARS-CoV-2 IgM (A, B) and SARS-CoV-2 IgG (C, D) were investigated. A sample was diluted with non-reactive serum in 5 to 8 steps; each sample was then analyzed with two replicates. The linearity studies for SARS-CoV-2 IgM (E) and SARS-CoV-2 IgG (F) were performed according to CLSI EP06-A. The sample of high antibody titer was diluted with the sample of low antibody titer in 6 to 9 steps; each sample was the analyzed with two steps.

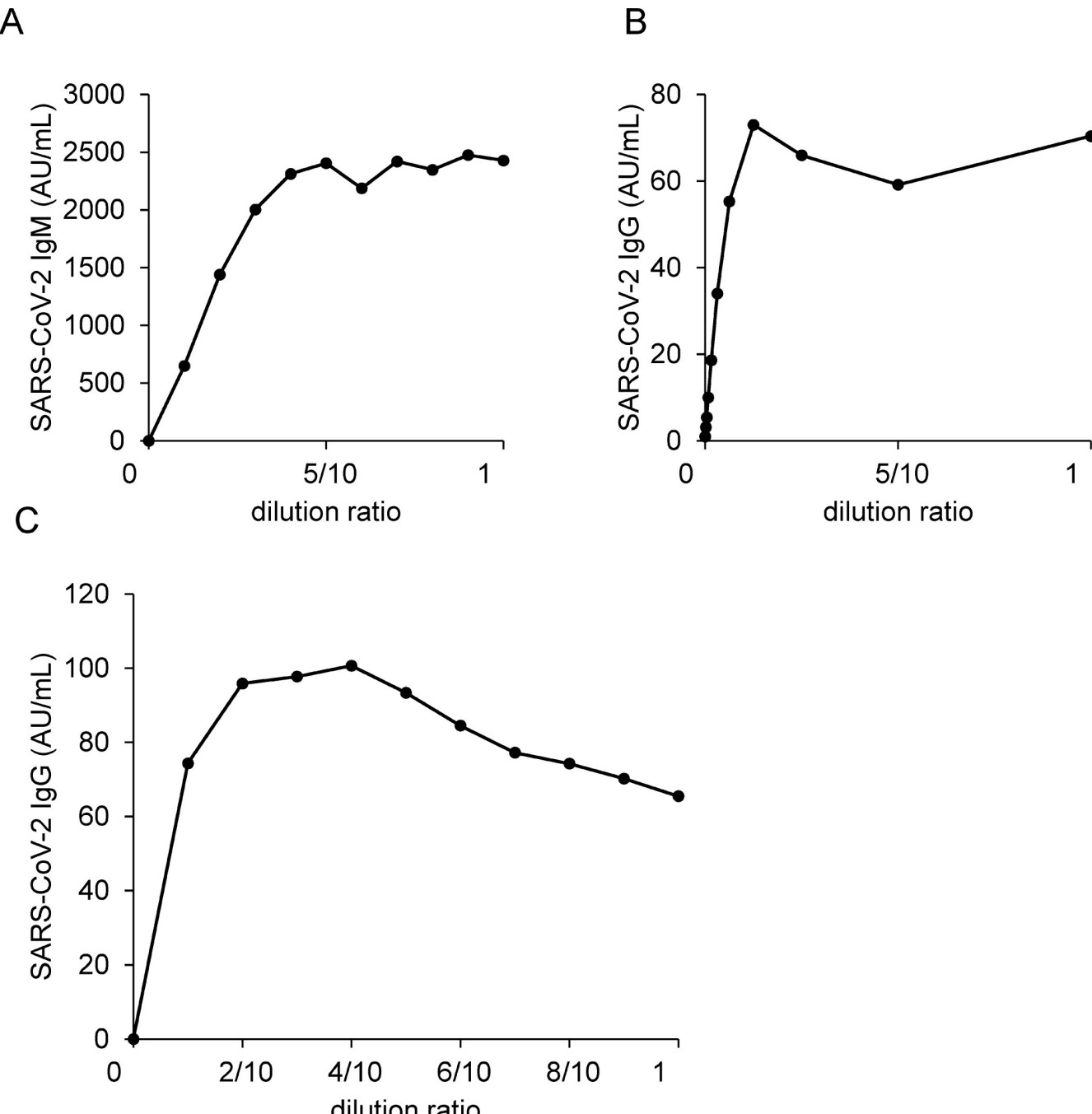

**Fig 2. Prozone phenomena and a hook effect of SARS-CoV-2 antibody titer.** The prozone phenomena of SARS-CoV-2 IgM (A) and SARS-CoV-2 IgG (B) were investigated. We diluted two serum samples from infected patients with non-reactive serum in 10 steps; each sample was then analyzed with two replicates. We diluted a serum sample with a saline in 10 steps and measured SARS-CoV-2 IgG titer (C); each sample was then analyzed with two replicates.

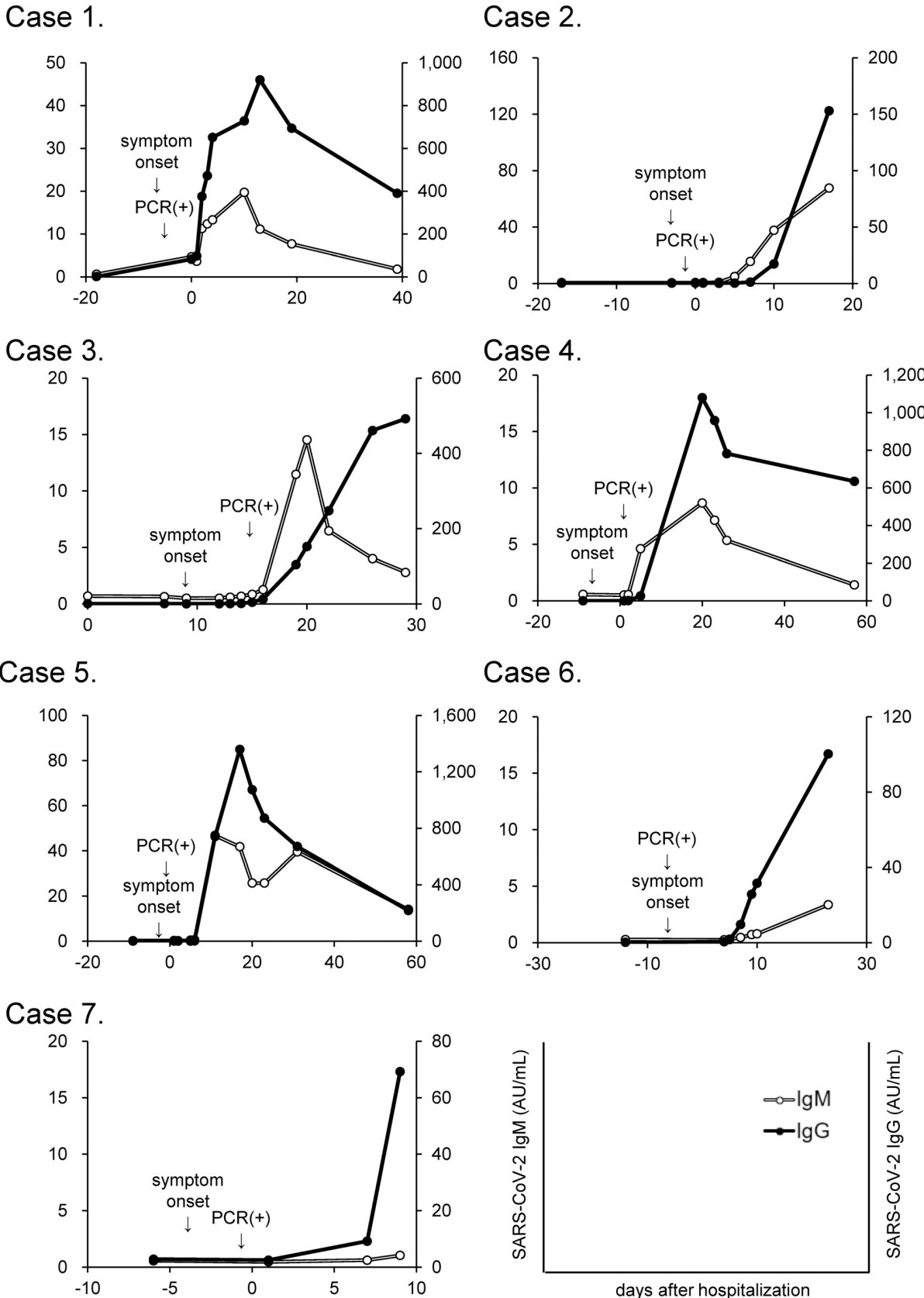

**Fig 3. Time course of serum antibody titers in COVID-19 subjects.** The time courses of the SARS-CoV-2 IgM and SARS-CoV-2 IgG titers in sera collected before and after the onset of COVID-19 were examined in seven patients.

Second, we obtained 100 random serum samples collected from outpatients who had visited The University of Tokyo Hospital in 2017, when SARS-CoV-2 did not exist. None of these samples had an antibody titer over 10 AU/mL, suggesting that this measurement system can detect SARS-CoV-2 IgM/IgG without false-positive results (Fig 4).

**Cross-reactivity with autoantibodies.** Since the presence of autoantibodies can sometimes affect the results of serological tests, we measured SARS-CoV-2 IgM/IgG in residual serum samples collected from patients with one of 5 different autoimmune diseases. For IgM, most of the serum samples from the patients with autoimmune diseases did not have a result over 10 AU/mL. However, two rheumatoid factor-positive patients, one anti-double-strand DNA antibody-positive patient, and one anti-mitochondrial M2 antibody-positive patient had values that exceeded the cutoff value (Fig 5A). For IgG, none of the autoantibody-positive serum samples had a result that was over 10 AU/mL (Fig 5B).

**Concordance rate with PCR testing or Roche's ECLIA kit.** To investigate clinical usefulness, we compared the results of the serological antibody tests with those of PCR tests. Among the 26 PCR-positive COVID-19 cases, 8 cases (30.8%) had IgM-positive results and 19 cases (73.1%) had IgG-positive results. Among the 53 PCR-negative COVID-19-suspected cases, 52 cases (98.1%) had IgM levels below 10 AU/mL and all the cases (100%) had IgG levels below 10 AU/mL (Table 2).

We also compared these results of the SARS-CoV-2 IgM or IgG tests with those obtained using the Roche's ECLIA kit. Among the 13 serum samples above 1 COI by the Roche's kit, 4 samples (30.8%) had the SARS-CoV-2 IgM-positive results and 12 samples (92.3%) had the SARS-CoV-2 IgG-positive results. Among the 48 serum samples under 1 COI by the Roche's

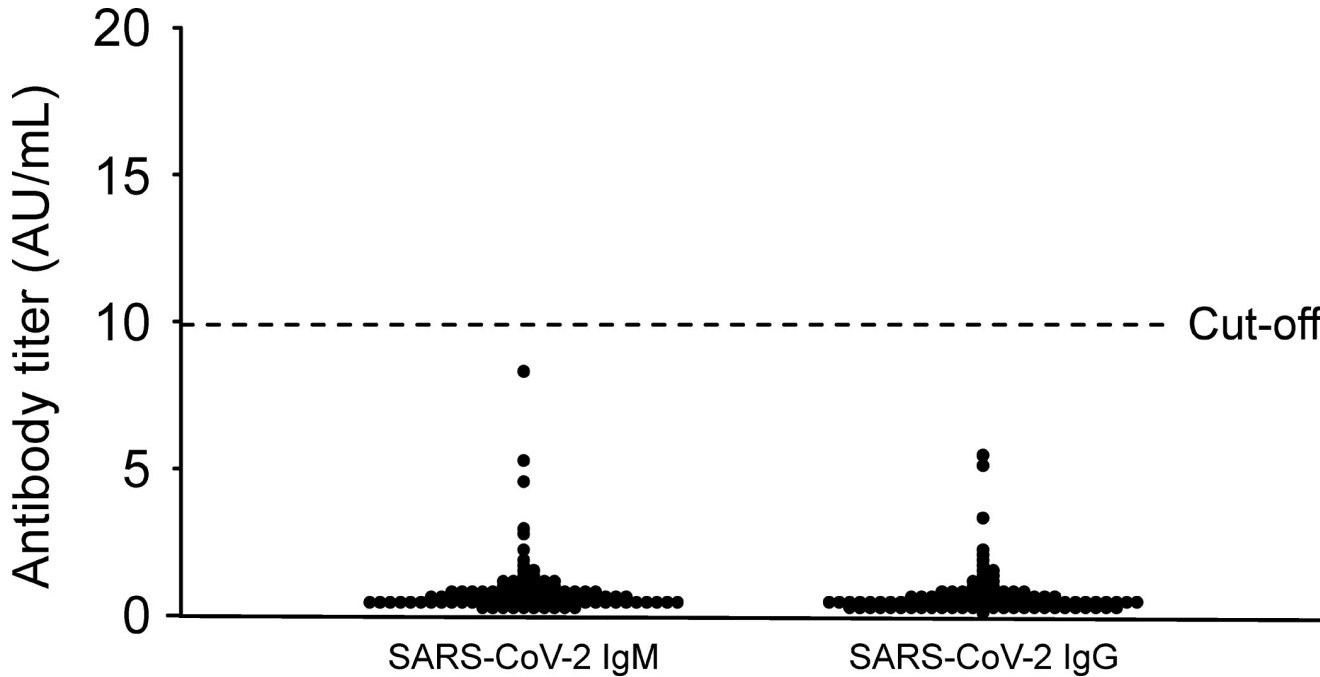

**Fig 4. Serum antibody titers in sera from 2017.** The SARS-CoV-2 IgM/IgG titers of sera collected from subjects (n = 100) in 2017 were measured.

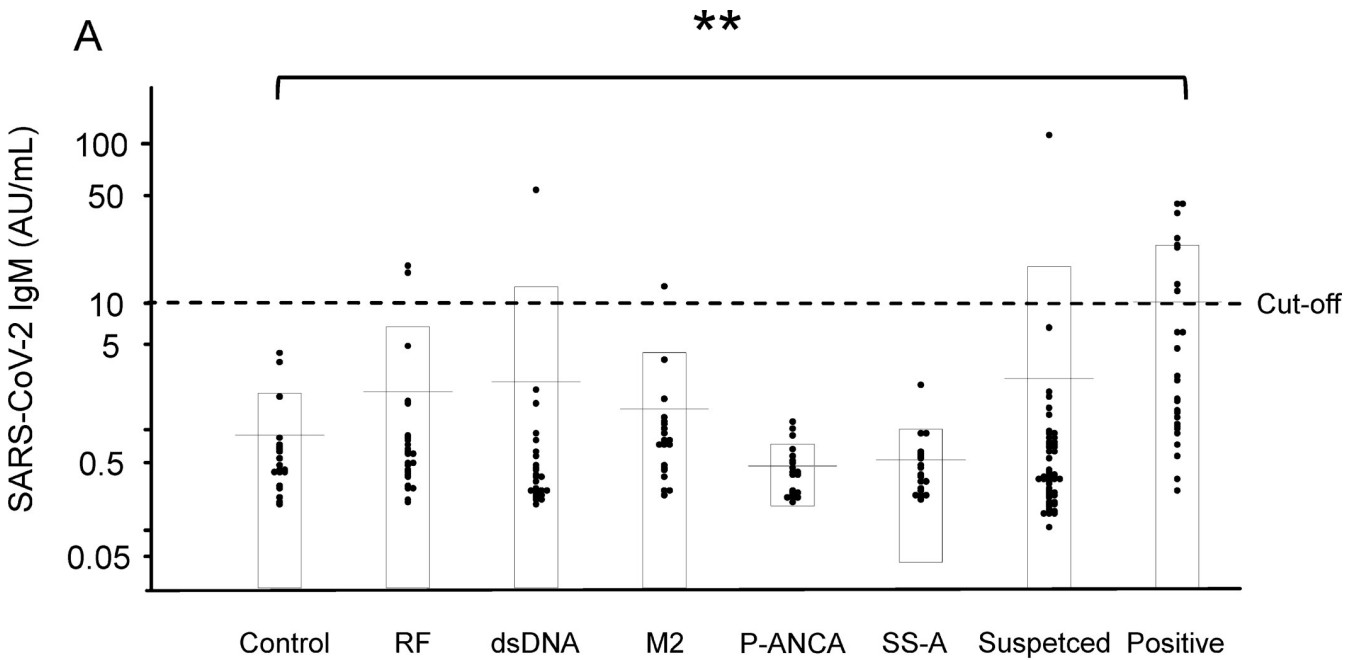

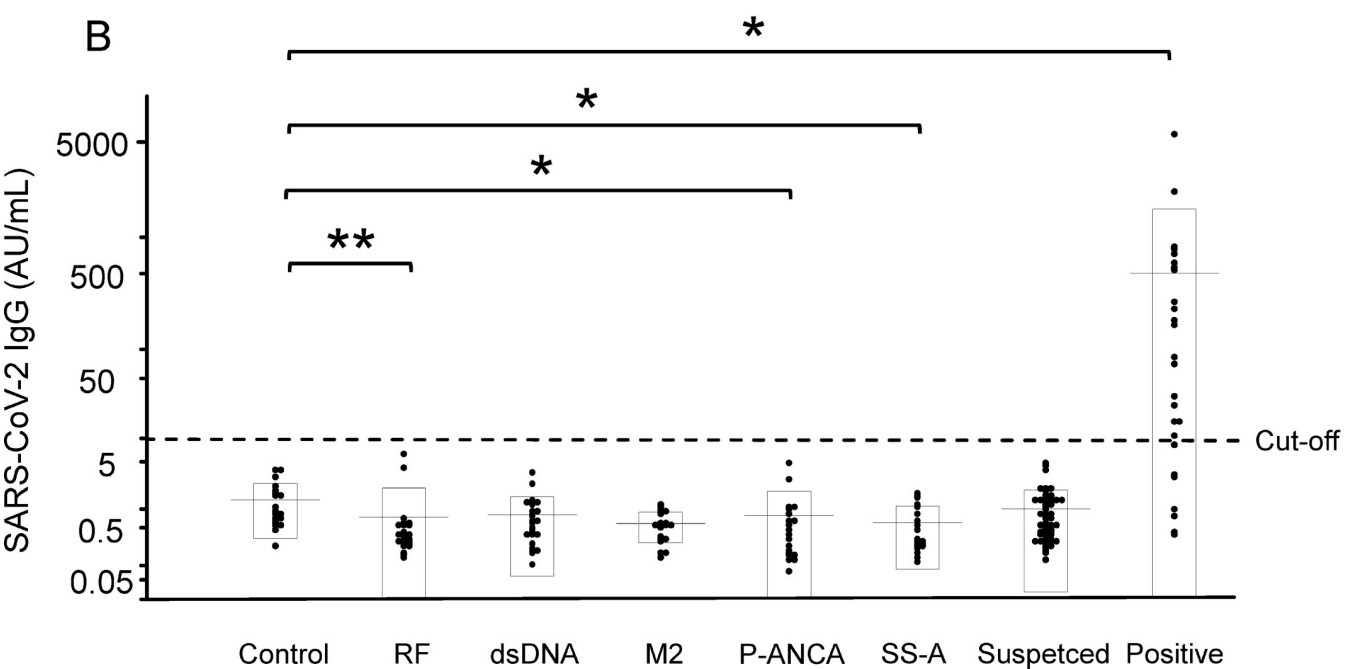

**Fig 5. Interference from autoantibodies in SARS-CoV-2 IgM/IgG assay.** We collected sera from patients with autoimmune diseases and measured the SARS-CoV-2 IgM (A) and IgG (B) titers. $*p < 0.05$, $**p < 0.01$. Control, randomly selected outpatients who visited the hospital in 2020 (n = 20); RF, rheumatoid factor-positive group (n = 25); dsDNA, anti-double-strand DNA antibody-positive group (n = 26); M2, anti-mitochondrial M2 antibody-positive group (n = 20); P-ANCA, myeloperoxidase antineutrophil cytoplasmic antibody-positive group (n = 20); SS-A, anti-Sjögren's syndrome A antibody-positive group (n = 20); Suspected, suspected COVID-19 group (n = 53); Positive, COVID-19-positive group (n = 26).

**Table 2. Concordance rate between the results of PCR testing and SARS-CoV-2 IgM or IgG serological testing.**

|  |  | SARS-CoV-2 PCR test | |
| --- | --- | --- | --- |
|  |  | positive cases (n = 26) | negative cases (n = 53) |
| SARS-CoV-2 IgM | >10 AU/mL | 8 (30.8%) | 1 |
|  | ≤10 AU/mL | 18 | 52 (98.1%) |
|  |  | SARS-CoV-2 PCR test | |
|  |  | positive cases (n = 26) | negative cases (n = 53) |
| SARS-CoV-2 IgG | >10 AU/mL | 19 (73.1%) | 0 |
|  | ≤10 AU/mL | 7 | 53 (100%) |

We measured SARS-CoV-2 IgM and IgG in PCR-positive subjects (n = 26) and PCR-negative subjects (n = 53). An antibody titer above 10 AU/mL was regarded as positive, according to the manufacturer's cutoff.

kit, 46 samples (95.8%) had the SARS-CoV-2 IgM or IgG levels below 10 AU/mL, when we analyzed only the samples identical to the samples in Table 2 (Table 3A). Among the 196 serum samples above 1 COI by the Roche's kit, 114 samples (58.2%) had the SARS-CoV-2 IgM-positive results and 193 samples (98.5%) had the SARS-CoV-2 IgG-positive results. Among the 189 serum samples under 1 COI, 179 samples (94.7%) had the SARS-CoV-2 IgM levels below 10 AU/mL and 165 samples (87.3%) had the SARS-CoV-2 IgG levels below 10 AU/mL, when we analyzed all the serum samples obtained additionally (Table 3B).

**Suspected false-positive IgM results might be caused by reactivity to N protein.** In this study, we observed 5 suspected false-positive IgM results. As described in the *Materials and Methods* section, the measurement system tests the reactivity to both the N protein and the S

**Table 3. Concordance rate between the results of the Roche's ECLIA kit and those of the SARS-CoV-2 IgM or IgG CLIA testing.**

**(A)**

|  |  | Roche's ECLIA kit | |
| --- | --- | --- | --- |
|  |  | ≥1 COI (n = 13) | <1 COI (n = 48) |
| SARS-CoV-2 IgM | ≥10 AU/mL | 4 (30.8%) | 2 |
|  | <10 AU/mL | 9 | 46 (95.8%) |
|  |  | Roche's ECLIA kit | |
|  |  | ≥1 COI (n = 13) | <1 COI (n = 48) |
| SARS-CoV-2 IgG | ≥10 AU/mL | 12 (92.3%) | 2 |
|  | <10 AU/mL | 1 | 46 (95.8%) |

**(B)**

|  |  | Roche's ECLIA kit | |
| --- | --- | --- | --- |
|  |  | ≥1 COI (n = 196) | <1 COI (n = 189) |
| SARS-CoV-2 IgM | ≥10 AU/mL | 114 (58.2%) | 10 |
|  | <10 AU/mL | 82 | 179 (94.7%) |
|  |  | Roche's ECLIA kit | |
|  |  | ≥1 COI (n = 196) | <1 COI (n = 189) |
| SARS-CoV-2 IgG | ≥10 AU/mL | 193 (98.5%) | 24 |
|  | <10 AU/mL | 3 | 165 (87.3%) |

We investigated the concordance rate between the SARS-CoV-2 IgM or IgG and the Roche's ECLIA kit in 61 serum samples used in Table 2 (A) and all of 385 serum samples collected from COVID-19-positive or -suspected patients (B). For the Roche's ECLIA kit, the SARS-CoV-2 antibody titer above 1 COI was regarded as positive, according to the manufacturer's cutoff.

**Table 4. IgM reactivity to N protein and S protein in subjects with suspected false-negative IgM results.**

| Sample | S+N protein (AU/mL) | N protein (AU/mL) | S protein (AU/mL) |
| --- | --- | --- | --- |
| 1 | 53.76 | 97.17 | 0.34 |
| 2 | 16.43 | 18.14 | 0.96 |
| 3 | 13.03 | 10.51 | 0.34 |
| 4 | 17.94 | 29.31 | 0.69 |
| 5 | 110.11 | 130.57 | 1.15 |

We investigated the IgM reactivity to N protein and S protein in five subjects with suspected false-negative IgM results. N or S protein means the value measured using magnetic beads coated with either antigen, respectively.

protein of SARS-CoV-2. We investigated the reactivity of the samples to N protein and S protein separately, and only reactivity to N protein was observed in the 5 suspected false-positive samples (Table 4).

## Discussion

In this study, we validated a method for quantifying SARS-CoV-2 IgM and IgG using the iFlash3000 automated CLIA analyzer. First, the repeatability and within-laboratory precision of the SARS-CoV-2 IgM and IgG assays were obviously good in validations, following to the CLSI document EP15-A3 (Table 1). These results suggest that relatively stable data were provided by this measurement system, while some anti-SARS-CoV-2 antibody detection kits authorized for emergency use by the Food and Drug Administration (FDA) in the USA have a CV of more than 5% for intermediate precision and some manufacturers do not even publish precision data for their products. The linearity was good up to values of 112.7 AU/mL for SARS-CoV-2 IgM and 55.3 AU/mL for IgG, while the assay signal gradually reached a plateau at over 2,405 AU/mL for SARS-CoV-2 IgM and over 73 AU/mL for IgG (Fig 2). Therefore, the sample should be diluted if the measured value is expected to be higher than the measurement range. To dilute samples, non-reactive serum should be used as a diluent, since a hook effect was observed when saline is used as a diluent, especially for IgG measurements (Fig 2C). Clinical decisions are rarely affected by this phenomenon, since the measurement range that was validated in the present study covers much higher values than the manufacturer's cutoff. Additionally, it would not be necessary to consider false negative results of SARS-CoV-2 IgG due to the extremely high titers, since the IgG titers of any participants who admitted to The University of Tokyo Hospital for COVID-19 had never become negative for SARS-CoV-2 IgG testing during the hospitalization once the seroconversion of IgG was observed. In case, the simultaneous measurement of IgM would help us rule out the possible false negativity in IgG test. To evaluate usefulness of these kits, we investigated the concordance rate between the results of the present study and those we obtained when we measured the same serum samples with Roche's ECLIA kit, as an approved method for the SARS-CoV-2 antibody testing. The concordance rate between SARS-CoV-2 IgM/IgG and Roche's kit were more than 90% except for the positive concordance rate between SARS-CoV-2 IgM and Roche's kit, when we analyzed only the same samples as described on Table 2 (Table 3A). The negative concordance rate between Roche's kit and SARS-CoV-2 IgM was 94.7% and the positive concordance rate between Roche's kit and SARS-CoV-2 IgG was 98.5%, when we compared using all the serum samples (Table 3B).

Regarding the samples showing the discrepant results between Roche's kit and SARS-CoV-2 IgG, three samples of which the antibody titers were below 10 AU/mL by SARS-CoV-2 IgG assay and above 1 COI by Roche's kit were PCR-positive subjects, while we also observed

seroconversion by SARS-CoV-2 IgG assay in the samples collected from the three subjects after 2 days. The antibody titers of 24 samples were above 10 AU/mL by SARS-CoV-2 IgG assay and below 1 COI by Roche's kit. Among these samples, 12 cases were PCR-positive cases. In eight of these cases, the seroconversions were observed in SARS-CoV-2 IgG assay at earlier point than Roche's kit, while, in other four cases, the antibody titers measured by Roche's kit were below 1 COI although those at the former or the latter points were above 1 COI. These results suggest that false-negative cases in the CLIA assay for SARS-CoV-2 IgG might be fewer than Roche's kit. According to the manufacturer's instruction, the Roche's kit adopts N protein as antigen to detect antibody and a principle of double antigen sandwich format. Therefore, the Roche's kit theoretically detects all classes of antibodies against the antigen immobilized on beads [24]. Therefore, the small discrepancy in the concordance rate between Roche's kit and SARS-CoV-2 IgM/IgG, especially IgM, may be derived from the difference in the principles of assays.

Regarding the antibody tests, in addition to issues surrounding accuracy, the matter of false-positive cases has also been a concern. As shown in Fig 3, the SARS-CoV-2 IgM/IgG levels were negative before hospitalization in three cases, and these antibody levels subsequently became positive after symptom onset in all cases, suggesting that this serological test can surely detect antibodies against SARS-CoV-2. In addition, we demonstrated that this measurement system could detect SARS-CoV-2 IgM/IgG without any false-positive results by evaluating 100 serum samples collected in 2017, when SARS-CoV-2 did not exist. These results confirm that this measurement system might be able to detect antibodies against SARS-CoV-2 alone, without cross-reactivity with other coronavirus strains that cause 15%-29% of all common colds [25, 26]. Chemiluminescence immunoassays are known to be affected by autoantibodies, such as rheumatoid factor, relatively often [27]. In the present study, we investigated whether five kinds of autoantibodies might interfere with the measurement system and found that no false-positive results were observed for the IgG assays, while four false-positive cases were observed for the IgM assays. We also found one PCR-negative case with a SARS-CoV-2 IgM titer over the cutoff value. Actually, when we investigated the reactivity of N protein and S protein to the sera separately, we found that the false-positive cases were caused by reactivity to the N protein (Table 4). The reason for this reactivity remains unclear at present but might be due to the cross-reactivity of the measurement system or, since the structure of the N protein of SARS-CoV-2 is similar to that of other coronavirus strains, antibodies to a structure similar to the N protein of SARS-CoV-2 might actually exist.

Finally, we investigated the concordance rate between PCR and this measurement system. As shown in Table 2, although all the PCR-negative subjects had negative results except for one subject with a high IgM level, the PCR-positive subjects did not necessarily have positive results for IgM or IgG. This discrepancy might be due to the fact that we used the serum sample that had been collected on the day nearest to the day on which the PCR sample had been collected. In several cases, only a few days had passed since the onset of symptoms, and IgG and IgM are reportedly not detectable during the early phase of COVID-19 [28]. Therefore, the time course for the appearance of IgG and IgM must be investigated for the application of antibody tests in clinical practice. It is difficult to compare the results of a molecular test and those of a serological assay because of the difference between the kinetics of viral RNA and those of antibodies. However, these results might help us understand the usefulness of the antibody testing in the diagnosis of COVID-19.

A characteristic of the present method is that both the N protein and the S protein are used as antigens. Reportedly, the sensitivity and specificity might depend on the types of antigens that are recognized by the antibodies, and the antibody to S protein is more sensitive than the antibody to N protein [29–31]. Since the measurement system in the present study uses both

the S and N proteins, this system might provide a greater sensitivity and diagnostic ability than an antibody test using either the S protein or the N protein alone. In contrast, analyzing the reactivity to both proteins could increase the number of false-positive cases, as described above. Further studies on the clinical significances of antibodies to N protein and S protein might be necessary to conclude which is most appropriate: measuring the reactivity to both proteins or to each protein separately.

In conclusion, we have validated a measurement system for detecting IgM and IgG against SARS-CoV-2 using CLIA kits and have observed that this system had sufficient performance for its introduction into clinical laboratory testing. Moreover, the possibility of false-negative results, especially for IgG against SARS-CoV-2, was relatively low. In the future, this method might be helpful for clinical diagnosis, epidemiological investigations, and the development of vaccines.

## Supporting information

**S1 Data. This data set shows all the antibody data analyzed in the present study.** (XLSX)

## Acknowledgments

We are thankful to the Murakami Foundation for the donation of the iFlash3000 to The University of Tokyo Hospital.

## Author Contributions

**Conceptualization:** Makoto Kurano, Chungen Qian, Fan He, Yoshiro Kishi, Jun Okada, Hitoshi Okazaki, Kyoji Moriya, Yasuyuki Seto, Tatsuhiko Kodama, Yutaka Yatomi.

**Data curation:** Naoyuki Yoshikawa, Yutaka Nagura.

**Formal analysis:** Rin Yokoyama.

**Investigation:** Makoto Kurano, Yoshifumi Morita, Takuya Shimura, Yuki Nakano, Chungen Qian, Fuzhen Xia, Fan He, Yoshiro Kishi, Jun Okada, Naoyuki Yoshikawa, Yutaka Nagura.

**Methodology:** Rin Yokoyama, Yoshifumi Morita, Takuya Shimura, Yuki Nakano, Chungen Qian, Fuzhen Xia, Fan He, Yoshiro Kishi, Jun Okada, Naoyuki Yoshikawa, Yutaka Nagura.

**Project administration:** Makoto Kurano.

**Supervision:** Makoto Kurano, Tatsuhiko Kodama, Yutaka Yatomi.

**Validation:** Rin Yokoyama.

**Writing – original draft:** Rin Yokoyama.

**Writing – review & editing:** Makoto Kurano, Chungen Qian, Fuzhen Xia, Fan He, Yoshiro Kishi, Jun Okada, Hitoshi Okazaki, Kyoji Moriya, Yasuyuki Seto, Tatsuhiko Kodama, Yutaka Yatomi.

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
