## [Decision Letter · Decision Letter 0]

4 Sep 2020

PONE-D-20-21741

Validation of a new automated chemiluminescent anti-SARS-CoV-2 IgM and IgG antibody assay system detecting both N and S proteins in Japan

PLOS ONE

Dear Dr. Kurano,

Thank you for submitting your manuscript to PLOS ONE. After careful consideration, we feel that it has merit but does not fully meet PLOS ONE’s publication criteria as it currently stands. Therefore, we invite you to submit a revised version of the manuscript that addresses the points raised during the review process.

In addition to the issues raised by the reviewer I would like to note the following:

1) To formally evaluate and do a verification of the analytical and clinical performance of a quantitative test specific guidelines need to be followed: e.g. CLSI EP15-A3. The authors' conclusion that the performance of the system is sufficient is not substantiated by sufficient data.

2) How was the number of the samples chosen?

3) Basic assay background details are missing, e.g. chemiluminescence details (dye?), volumes used, calibrators, correlation between light units and AU antibody/ml.

4) Typically a new method needs to be compared to a gold standard- maybe a validated ELISA in this case? PCR is not an appropriate comparison for this assay.

5) How was serum collected and processed?

6) No formal estimation of LOB and LOD, LOQ  has been presented.

7) Linearity studies should be performed according to CLSI EP-06.

8) Figure 3- how were the "pre-infection" samples  in the 3 patients collected?

9) Line 115- typo "coLLected"

10) A quick search shows that the YHLO chemiluminescent anti-SARS-CoV-2 Ab assay is under FDA EUA and CE marked.

We look forward to receiving your revised manuscript.

Kind regards,

Katerina Kourentzi, PhD

Academic Editor

PLOS ONE

Journal Requirements:

2. Please clarify whether all samples used in this study were stored biological samples, and whether samples were de-identified before researchers accessed them.

In addition, please specify whether your IRB specifically approved your informed consent plan (opt-out on website).

3. Thank you for including your competing interests statement; "The present study is a collaborative research project among The University of Tokyo, Shenzhen YHLO Biotech Co., Ltd, and Medical & Biological Laboratories Co., Ltd. F. X. and F. H. are employees of Shenzhen YHLO Biotech Co., Ltd and Y. K. and J. O. are employees of Medical & Biological Laboratories Co., Ltd."

We note that one or more of the authors are employed by a commercial company: Shenzhen YHLO Biotech Co., Ltd, Medical & Biological Laboratories Co., Ltd.

Reviewers' comments:

Reviewer's Responses to Questions

**Comments to the Author**

1. Is the manuscript technically sound, and do the data support the conclusions?

Reviewer #1: Yes

2. Has the statistical analysis been performed appropriately and rigorously? 

Reviewer #1: Yes

3. Have the authors made all data underlying the findings in their manuscript fully available?

Reviewer #1: Yes

4. Is the manuscript presented in an intelligible fashion and written in standard English?

Reviewer #1: No

5. Review Comments to the Author

Reviewer #1: The study lacks originality. Even if the new should be linked to the application to a Japonase population , this is not enough. I suggest to make it as concise report or letter and submit to a different journal.

Some techinical details should be better explained in the text such as how they tested the antibodies against the single antigen (N or S) and why they used the non reactive serum to diluite the positive serum . The kit istruction don't say this and it's a strong action since they made linearity studies.

6. PLOS authors have the option to publish the peer review history of their article (what does this mean?). If published, this will include your full peer review and any attached files.

Reviewer #1: No

---

## [Author Response · Author response to Decision Letter 0]

29 Oct 2020

Response to the Review Comments to the Author

Reviewer #1: The study lacks originality. Even if the new should be linked to the application to a Japonase population , this is not enough. I suggest to make it as concise report or letter and submit to a different journal.

Some techinical details should be better explained in the text such as how they tested the antibodies against the single antigen (N or S) and why they used the non reactive serum to diluite the positive serum . The kit istruction don't say this and it's a strong action since they made linearity studies.

 We appreciate the critical point raised from Reviewer #1. We should admit that many articles have investigated the validity and usefulness of antibody tests, however since the clinical features of COVID-19 vary among countries, we believe it is an important task to investigate the validity and utility of antibody tests in Japan, where few researches have investigated those of antibody tests. 

 We apologize for the insufficient description on how to measure the antibody titers against the single antigen. We measured those antibody titers using magnetic beads coated with either antigen, respectively. The basic assay procedure is the same as the method of the original kit. We added this point to the Material and Method section in the revised manuscript (line 144 – 145).

 We used non-reactive serum to dilute the positive serum, since a hook effect was observed in the measurement of SARS-CoV-2 IgG when we dilute the samples with saline. We added this result to the revised manuscript (line 233 – 236, Figure 2C).

Response to the Editor’s comments’

In addition to the issues raised by the reviewer I would like to note the following:

1) To formally evaluate and do a verification of the analytical and clinical performance of a quantitative test specific guidelines need to be followed: e.g. CLSI EP15-A3. The authors' conclusion that the performance of the system is sufficient is not substantiated by sufficient data.

 We appreciate this suggestion by the Editor. We performed the verification of precision, using two kinds of samples each assay for 5 days, 5 replicates per run, according to the CLSI document EP15-A3. 

Repeatability and within-laboratory precision of all samples were lower than 5% and the upper verification limits described in the original data published by the manufacture (Ref. 20). We added these results to the revised manuscript (line 197 – 207 and Table 2).

2) How was the number of the samples chosen?

 We apologize for the insufficient description on how to choose the number of the samples. We enrolled the COVID-19 subjects and the suspicious subjects, which we can collect their residual serum samples between April 22 and June 22. We added this point in the revised manuscript (line 118 – 119)

3) Basic assay background details are missing, e.g. chemiluminescence details (dye?), volumes used, calibrators, correlation between light units and AU antibody/ml.

 We apologize for the insufficient description on the assay information. These assays required 5 uL of the samples and the acridinium-labeled anti-human IgM or IgG conjugate antibody was used to detect them. The unites “AU/mL” are determined by comparing the RLU detected by iFlash optical system with the cutoff calculated from the SARS-CoV-2 IgM or IgG calibrators containing anti-SARS-CoV-2 IgM or IgG chimeric antibody. We added this information to the revised manuscript (line 136 – 142).

4) Typically a new method needs to be compared to a gold standard- maybe a validated ELISA in this case? PCR is not an appropriate comparison for this assay. 

 We appreciate this point raised from Editor. Unfortunately, because of the biosafety facility in our laboratory, we are unable to perform a manual-based ELISA to measure the antibody titers of COVID-19 patient. Instead of ELISA, we compared the chemiluminescent anti-SARS-CoV-2 antibody tests with the Roche’s COVID-19 antibody test, which received FDA emergency use authorization and was available in markets accepting the CD mark. According to the manufacturer’s instruction of the Roche’s test, the cutoff value for a positive COVID-19 antibody result was deemed as 1.0 cutoff index (COI). The positive and negative concordance rates between Roche’s test and SARS-CoV-2 IgM or IgG serological testing showed more than 85%, except for the positive concordance rate of SARS-CoV-2 IgM. We added these results to the Method section, the Result section, and the Discussion section in the revised manuscript (line 148 – 153, line 310 – 315, and line 360 – 371 and Table 4). 

5) How was serum collected and processed?

 We apologize for the lack of the description on how to collect serum. We collected the sera in the following steps. We received the patient’s whole blood in the collection tube coated with silica and thrombin for the clinical laboratory testing. Then, the serum was separated by centrifuging at 2,300 g for 5 minutes and carried out clinical routine testing. Subsequently, we collected and stored the residual samples at -20℃ until the antibody testing. We added these pointes in the revised manuscript (line112 – 115).

6) No formal estimation of LOB and LOD, LOQ has been presented.

 We appreciate this comment from the Editor. As suggested by Editor, we added the formal estimation of LoB, LoD and LoQ, according to the guidelines in CLSI document EP17. The LoD for SARS-CoV-2 IgM was 0.74 AU/mL, determined by 130 measurements with 60 blank and 70 low level replicates, and the LoB was 0.63 AU/mL. The LoD for SARS-CoV-2 IgG was 0.53 AU/mL, determined by 120 measurements with 60 blank and 60 low level replicates, and the LoB was 0.47 AU/mL. We also investigated the LoQ. We measured 7 samples of low antibody levels for SARS-CoV-2 IgM and 10 samples for SARS-CoV-2 IgG. In both assays, CVs of the samples which were lower than the LoD showed no more than 10%. Therefore, the LoQ for SARS-CoV-2 IgM was determined as 0.74 AU/mL and that for SARS-CoV-2 IgG was 0.53 AU/mL. We added those results in the revised manuscript (line 175 – 182 and line 253 – 262).

7) Linearity studies should be performed according to CLSI EP-06.

 We appreciate this comment from the Editor. As suggested by Editor, we performed the linearity studies, following the guidelines in CLSI document EP06. For IgM, the method was demonstrated to be linear between 1.50 and 15.92 AU/mL since any nonlinear coefficients were not significant in this range. And for IgG, the linear range was between 2.36 and 18.30 AU/mL since any nonlinear coefficients were not significant in this range. We added these results in the revised manuscript (line 225 – 229 and Figure 1E, F).

8) Figure 3- how were the "pre-infection" samples in the 3 patients collected?

 We apologize for the insufficient information on how to collect the “pre-infection” samples. In our laboratory, the sera which completed clinical laboratory test were usually stored at -20℃ up to 3 weeks. We collected these samples since the sera which were collected and stored in our laboratory before the onset of COVID-19 symptom by chance. We added those points in the revised manuscript (line 115 – 118).

9) Line 115- typo "collected

"

 We apologize for a misspelling. We have corrected this part. Thank you very much for kind suggestion (line 123).

10) A quick search shows that the YHLO chemiluminescent anti-SARS-CoV-2 Ab assay is under FDA EUA and CE marked.

 We appreciate this kind information. Exactly, the YHLO chemiluminescent anti-SARS-Cov-2 Antibody assay acquired these authorizations. We believe that the evaluation in academic organizations is important to establish the clinical usefulness of these assays.

Response to the Journal Requirement

1. Please clarify whether all samples used in this study were stored biological samples, and whether samples were de-identified before researchers accessed them.

 In addition, please specify whether your IRB specifically approved your informed consent plan (opt-out on website).

 We have clarified these points on line 124 – 125 and line 127 – 128 in the revised manuscript.

 2. Thank you for including your competing interests statement; "The present study is a collaborative research project among The University of Tokyo, Shenzhen YHLO Biotech Co., Ltd, and Medical & Biological Laboratories Co., Ltd. F. X. and F. H. are employees of Shenzhen YHLO Biotech Co., Ltd and Y. K. and J. O. are employees of Medical & Biological Laboratories Co., Ltd."

 We note that one or more of the authors are employed by a commercial company: Shenzhen YHLO Biotech Co., Ltd, Medical & Biological Laboratories Co., Ltd.

 We have added the commercial affiliation in the section of funding sources (line 426 – 427) and included the requested statement (line 428 – 431).

 We also revised the author contributions section (line 436 – 438) in more detail and update author roles in the online submission form.

 As requested, we revised the Competing Interests Statement (line 446 – 447) and added the requested statement (line 448 – 451).

 3. Thank you for updating your data availability statement. You note that your data are available within the Supporting Information files, but no such files have been included with your submission. At this time we ask that you please upload your minimal data set as a Supporting Information file, or to a public repository such as Figshare or Dryad. 

 Please also ensure that when you upload your file you include separate captions for your supplementary files at the end of your manuscript.

 As soon as you confirm the location of the data underlying your findings, we will be able to proceed with the review of your submission.

 We apologized not to include the supporting information files for the data set. We submitted the file.

---

## [Decision Letter · Decision Letter 1]

8 Dec 2020

PONE-D-20-21741R1

Validation of a new automated chemiluminescent anti-SARS-CoV-2 IgM and IgG antibody assay system detecting both N and S proteins in Japan

PLOS ONE

Dear Dr. Kurano,

Thank you for submitting your revised manuscript to PLOS ONE. After careful consideration, we feel that it has merit but does not fully meet PLOS ONE’s publication criteria as it currently stands. Therefore, we invite you to submit a revised version of the manuscript that addresses the points raised during the review process.

A rebuttal letter that responds to each point raised by the academic editor and reviewer. You should upload this letter as a separate file labeled 'Response to Reviewers'.A marked-up copy of your manuscript that highlights changes made to the original version. You should upload this as a separate file labeled 'Revised Manuscript with Track Changes'.An unmarked version of your revised paper without tracked changes. You should upload this as a separate file labeled 'Manuscript'.

We look forward to receiving your revised manuscript.

Kind regards,

Katerina Kourentzi, PhD

Academic Editor

PLOS ONE

Reviewers' comments:

Reviewer's Responses to Questions

**Comments to the Author**

1. If the authors have adequately addressed your comments raised in a previous round of review and you feel that this manuscript is now acceptable for publication, you may indicate that here to bypass the “Comments to the Author” section, enter your conflict of interest statement in the “Confidential to Editor” section, and submit your "Accept" recommendation.

Reviewer #2: All comments have been addressed

2. Is the manuscript technically sound, and do the data support the conclusions?

Reviewer #2: Partly

3. Has the statistical analysis been performed appropriately and rigorously? 

Reviewer #2: Yes

4. Have the authors made all data underlying the findings in their manuscript fully available?

Reviewer #2: Yes

5. Is the manuscript presented in an intelligible fashion and written in standard English?

Reviewer #2: Yes

6. Review Comments to the Author

Reviewer #2: The study of Yokoyama and colleagues is about a validation of a chemiluminescent assay for detection of antibodies IgM and IgG against SARS-CoV-2. Although there is a big importance in the validation of new methods to help in Covid-19 diagnosis, the study has some deficiencies regarding the study design.

1) To assess the applicability of the method using real patient samples, serum samples from 26 participants with confirmed PCR and 53 from suspect cases without laboratory confirmation were included. I believe that a larger and better characterized sample panel should have been used for this evaluation. Furthermore, authors should inform readers about the demographic and clinical profile of such participants, as well as for the negative controls (collected before the period and with autoimmune diseases) in a table.

2) Still regarding the positive participants samples, analysis of the sensitivity of the assay was made according to the PCR result and the authors only give the information that serum samples were collected a few days after PCR confirmation. What is the mean number of days of collection post symptoms onset? The first reviewer also mentioned the difficulty of compare a molecular and a serological assay because of the RNA and antibodies dynamics and suggested the use of a commercial ELISA method, which was incorporated to the analysis.

3) It is not clear for me why in the comparison of CLIA and the assay from Roche, much more positive serum samples were used. Is it possible to access PCR results from these individuals? Please include more characteristics of the samples and participants.

4) Authors explain that samples collected before the onset of the infection were collected by chance. Did you have performed any laboratorial detection assay to guarantee that these subjects did not have current or past SARS-CoV-2 infection at that time? The same question can be applied for negative controls collected on March 2020.

5) For the specificity analysis of the CLIA assay, is it possible to include samples from participants with confirmed coronavirus otherwise than SARS-COV-2 and also with other respiratory infections? The analysis of specificity with autoantibodies was very important.

6) The method validation is well described and after first revision, it was included a validation according to a protocol (EP15-A3). Authors should rethink if it is necessary both analysis or only the validated one.

7) Please include citation of Table 2A in line 196.

8) It is not clear for me how future researchers will interpret a negative result on CLIA as a possible situation of high titers of antibodies and false negativity. How can this work on the routine clinical practice?

9) In figure 3, authors showed the IgM and IgG dynamics before and after PCR confirmation. I’m not sure if data of only three participants are strong for this analysis. Furthermore, how do you explain the low titers of IgM in 2 of 3 of these participants?

10) The concordance of CLIA assay and Roche assay was not very good mainly for IgG detection. How do you interpret this divergence? So, I don’t agree with the sentence in the discussion lines 365 and 366.

11) Please include the new information of the study in the abstract section.

7. PLOS authors have the option to publish the peer review history of their article (what does this mean?). If published, this will include your full peer review and any attached files.

Reviewer #2: No

---

## [Author Response · Author response to Decision Letter 1]

15 Jan 2021

Point-by-Point Responses to the Reviewers’ Comments

For PONE-D-20-21741R1

#Correction

 First of all, we apologize for inappropriate interpretation in the previous version that the samples of which the titers were equal to the cutoff value had been regard as positive for SARS-CoV-2 IgM/IgG assay and Roche’s kit. In this revised manuscript, we calculated these results again and as a result one sample was actually positive for Roche’s kit. We revised Table 3 and corrected the numbers of samples and concordance rates in line 308-312.

Response to the Review Comments to the Author

Reviewer #2: The study of Yokoyama and colleagues is about a validation of a chemiluminescent assay for detection of antibodies IgM and IgG against SARS-CoV-2. Although there is a big importance in the validation of new methods to help in Covid-19 diagnosis, the study has some deficiencies regarding the study design.

1) To assess the applicability of the method using real patient samples, serum samples from 26 participants with confirmed PCR and 53 from suspect cases without laboratory confirmation were included. I believe that a larger and better characterized sample panel should have been used for this evaluation. Furthermore, authors should inform readers about the demographic and clinical profile of such participants, as well as for the negative controls (collected before the period and with autoimmune diseases) in a table.

The samples were collected from subjects with negative PCR results at the same day as the PCR test was performed and there were no subjects suspected of autoimmune diseases. For the outpatients who had visited The University of Tokyo Hospital, we have chosen the subjects without autoimmune diseases. We added this information in line 127. 

2) Still regarding the positive participants samples, analysis of the sensitivity of the assay was made according to the PCR result and the authors only give the information that serum samples were collected a few days after PCR confirmation. What is the mean number of days of collection post symptoms onset? The first reviewer also mentioned the difficulty of compare a molecular and a serological assay because of the RNA and antibodies dynamics and suggested the use of a commercial ELISA method, which was incorporated to the analysis.

The mean days (±S.D.) between the antibody test and the onset of the symptom or the PCR test were 11.3 (±6.70) or 5.67 (±5.67) days, respectively. We added this information in line 125-127.

Of course, we admit that the difficulty in comparing a molecular and a serological assay because of the RNA and antibodies dynamics. However, the aim of the present study is not only to validate our assays but also to investigate the usefulness in the diagnosis. Therefore, we compared the results of our assay with those of PCR tests. We added the difficulty in comparing both results in the revised manuscript (line 418-421).

Regarding the comparison with an ELISA, we compared out results with the results of FDA-approved assay from Roche, since manual based ELISAs have not been established to measure the antibodies against SARS-CoV-2 in Japan in addition to the difficulty in performing manual based ELISAs from the aspects of bio safety in our laboratory.

Judging from the results in the comparison with the Roche’s assay which measured antibodies without separating immunoglobulin subclasses, IgG test had a rather high concordance with the previous assay.

3) It is not clear for me why in the comparison of CLIA and the assay from Roche, much more positive serum samples were used. Is it possible to access PCR results from these individuals? Please include more characteristics of the samples and participants.

We apologize for the insufficient description about the characteristics of samples and participants we used in the comparison of CLIA and the assay from Roche. For this experiment (new Table 3B), we used all the serum samples which were collected from COVID-19-positive patient defined as the method section. In some cases, several serum samples collected from one patient were used. 

To avoid this concern, we added the results of the concordance rates when we limited the samples only to the same samples as used in Table 2 (new Table 3A).

4) Authors explain that samples collected before the onset of the infection were collected by chance. Did you have performed any laboratorial detection assay to guarantee that these subjects did not have current or past SARS-CoV-2 infection at that time? The same question can be applied for negative controls collected on March 2020.

Two of those subjects had been confirmed SARS-CoV-2 PCR negative before the onset of COVID-19 symptoms. In one subject and control cases collected in March 2020, symptoms of cold were not described in medical records. In the revised manuscript, we added four cases of whom the samples collected before the onset of COVID-19 were available. One of the additional cases was confirmed PCR-negative before the onset of COVID-19 symptoms and regarding other cases, symptoms of cold were not described in medical records. We added this detail information on the subjects in line 119-122 in the revised manuscript.

5) For the specificity analysis of the CLIA assay, is it possible to include samples from participants with confirmed coronavirus otherwise than SARS-COV-2 and also with other respiratory infections? The analysis of specificity with autoantibodies was very important.

We agree that this is important point. At present, we did not obtain the serum of subjects infected with other coronaviruses. Instead, the manufacturer had assessed the cross-reactivity using serum samples of the subjects who had been confirmed with other respiratory infection (please see FYI Figures). These results suggested that these antibody tests did not have cross-reactivities against coronavirus other than SARS-CoV-2, such as OC43 and HKU1. These raw data, however, had not been permitted to be published by Shenzhen YHLO Biotech Co., Ltd. Therefore, we did not include these data in this paper.

 6) The method validation is well described and after first revision, it was included a validation according to a protocol (EP15-A3). Authors should rethink if it is necessary both analysis or only the validated one.

 We appreciate this kind suggestion. We will include the results of validation according to the CLSI document EP15-A3 and deleted the other results.

7) Please include citation of Table 2A in line 196.

 We appreciate this kind suggestion. We added the citation in line 197 (new Table 1 in the revised manuscript). 

8) It is not clear for me how future researchers will interpret a negative result on CLIA as a possible situation of high titers of antibodies and false negativity. How can this work on the routine clinical practice?

 We appreciate the critical point raised from the Reviewer #2. Actually, the results on Figure 2C might propose the possibility that extremely high titers of IgG might result in false-negative result. However, to our experience, IgG titers of any participants who admitted to The University of Tokyo Hospital for COVID-19 (n = 83) had never become negative for SARS-CoV-2 IgG testing during the hospitalization once the seroconversion of IgG was observed. Therefore, we think that it would not be necessary to consider false negative results of IgG due to the extremely high titers and to dilute the serum samples of negative SARS-CoV-2 IgG results on the routine clinical practice. The simultaneous measurement of IgM would help the researchers to rule out the false negativity in IgG tests, since a hook effect was not observed in the SARS-CoV-2 IgM assay. We added this point in the Discussion section (line 360-365).

9) In figure 3, authors showed the IgM and IgG dynamics before and after PCR confirmation. I’m not sure if data of only three participants are strong for this analysis. Furthermore, how do you explain the low titers of IgM in 2 of 3 of these participants?

 In the revising the manuscript, we enrolled four more participants of whom serum samples before and after PCR conformation were available. These additional results showed that SARS-CoV-2 IgM/IgG titers were changed from the titers below the cut-off level to those over the cut-off value at all 7 participants. 

 Regarding the low levels of IgM titers, the titers become over cutoff value in 4 of 7 of those participants. We rearranged the scales of SARS-CoV-2 IgM and IgG titers to present the results clearly (Figure 3). We revised the manuscript (line 262-264).

10) The concordance of CLIA assay and Roche assay was not very good mainly for IgG detection. How do you interpret this divergence? So, I don’t agree with the sentence in the discussion lines 365 and 366.

 We apologize for the insufficient description about the concordance rate of CLIA assay and Roche’s kit. We added the detail of the samples of which the results were not concordant (line 374-383); three samples of which the antibody titers were below 10 AU/mL by SARS-CoV-2 IgG assay and above 1 COI by Roche’s kit were PCR-positive subjects, while we also observed seroconversion by SARS-CoV-2 IgG assay in the samples collected from the three subjects after 2 days. The antibody titers of 24 samples were above 10 AU/mL by SARS-CoV-2 IgG assay and below 1 COI by Roche’s kit. Among these samples, 12 cases were PCR-positive cases. In eight of these cases, the seroconversions were observed in SARS-CoV-2 IgG assay at earlier point than Roche’s kit, while, in other four cases, the antibody titers measured by Roche’s kit were below 1 COI although the antibody titers at the former or the latter points were above 1 COI. 

Thinking these points, false-negative cases in the CLIA assay for SARS-CoV-2 IgG were fewer than Roche’s assay. We added this point in the revised manuscript (line 383-384).

 Also, we deleted the description in line 365 and 366 in the previous manuscript according to the suggestion from the Reviewer #2.

11) Please include the new information of the study in the abstract section.

 We appreciate this comment from the Reviewer #2. As suggested from Reviewer #2, we added the new information in the abstract section (line 62-64).

---

## [Decision Letter · Decision Letter 2]

12 Feb 2021

Validation of a new automated chemiluminescent anti-SARS-CoV-2 IgM and IgG antibody assay system detecting both N and S proteins in Japan

PONE-D-20-21741R2

Dear Dr. Kurano,

We’re pleased to inform you that your manuscript has been judged scientifically suitable for publication and will be formally accepted for publication once it meets all outstanding technical requirements.

Kind regards,

Katerina Kourentzi, PhD

Academic Editor

PLOS ONE

Additional Editor Comments (optional):

Reviewers' comments:

Reviewer's Responses to Questions

**Comments to the Author**

1. If the authors have adequately addressed your comments raised in a previous round of review and you feel that this manuscript is now acceptable for publication, you may indicate that here to bypass the “Comments to the Author” section, enter your conflict of interest statement in the “Confidential to Editor” section, and submit your "Accept" recommendation.

Reviewer #2: All comments have been addressed

2. Is the manuscript technically sound, and do the data support the conclusions?

Reviewer #2: Yes

3. Has the statistical analysis been performed appropriately and rigorously? 

Reviewer #2: Yes

4. Have the authors made all data underlying the findings in their manuscript fully available?

Reviewer #2: Yes

5. Is the manuscript presented in an intelligible fashion and written in standard English?

Reviewer #2: Yes

6. Review Comments to the Author

Reviewer #2: (No Response)

7. PLOS authors have the option to publish the peer review history of their article (what does this mean?). If published, this will include your full peer review and any attached files.

Reviewer #2: No

---

## [Editor Report · Acceptance letter]

23 Feb 2021

PONE-D-20-21741R2 

Validation of a new automated chemiluminescent anti-SARS-CoV-2 IgM and IgG antibody assay system detecting both N and S proteins in Japan 

Dear Dr. Kurano:

I'm pleased to inform you that your manuscript has been deemed suitable for publication in PLOS ONE. Congratulations! Your manuscript is now with our production department. 

Kind regards, 

on behalf of

Dr. Katerina Kourentzi 

Academic Editor

PLOS ONE